# Targeting the pH Paradigm at the Bedside: A Practical Approach

**DOI:** 10.3390/ijms21239221

**Published:** 2020-12-03

**Authors:** Tomas Koltai

**Affiliations:** Centro de Diagnostico y Tratamiento de la Obra Social del Personal de la Alimentacion, Talar de Pacheco, Buenos Aires 1617, Argentina; tkoltai@hotmail.com

**Keywords:** pH paradigm, pHtome, amiloride, proton pump inhibitors, acetazolamide, topiramate, quercetin, repurposed drugs

## Abstract

The inversion of the pH gradient in malignant tumors, known as the pH paradigm, is increasingly becoming accepted by the scientific community as a hallmark of cancer. Accumulated evidence shows that this is not simply a metabolic consequence of a dysregulated behavior, but rather an essential process in the physiopathology of accelerated proliferation and invasion. From the over-simplification of increased lactate production as the cause of the paradigm, as initially proposed, basic science researchers have arrived at highly complex and far-reaching knowledge, that substantially modified that initial belief. These new developments show that the paradigm entails a different regulation of membrane transporters, electrolyte exchangers, cellular and membrane enzymes, water trafficking, specialized membrane structures, transcription factors, and metabolic changes that go far beyond fermentative glycolysis. This complex world of dysregulations is still shuttered behind the walls of experimental laboratories and has not yet reached bedside medicine. However, there are many known pharmaceuticals and nutraceuticals that are capable of targeting the pH paradigm. Most of these products are well known, have low toxicity, and are also inexpensive. They need to be repurposed, and this would entail shorter clinical studies and enormous cost savings if we compare them with the time and expense required for the development of a new molecule. Will targeting the pH paradigm solve the “cancer problem”? Absolutely not. However, reversing the pH inversion would strongly enhance standard treatments, rendering them more efficient, and in some cases permitting lower doses of toxic drugs. This article’s goal is to describe how to reverse the pH gradient inversion with existing drugs and nutraceuticals that can easily be used in bedside medicine, without adding toxicity to established treatments. It also aims at increasing awareness among practicing physicians that targeting the pH paradigm would be able to improve the results of standard therapies. Some clinical cases will be presented as well, showing how the pH gradient inversion can be treated at the bedside in a simple manner with repurposed drugs.

## 1. Introduction

Between 1980 and 2000, with the identification of protooncogenes, tumor suppressor genes, metabolic and signaling pathways and the many proteins involved in cancer’s development and progression, the general belief was that cancer is a chaotic process [1]. 

However, in 1996, Kinzler and Vogelstein [2] after many years of meticulous work described the step-by-step genetic evolution of colon cancer from adenoma to invasive disease. Thus, cancer was not so chaotic. In 2000, two distinguished researchers, worried by this vision of random development of malignancies, published their seminal paper “The Hallmarks of Cancer” [3]. This review showed for the first time that malignant tumors are quite well organized and follow certain basic rules. The authors, Hanahan and Weinberg, described six hallmarks that all cancer cells follow in order to become malignant and progress. In a further paper, in 2011, they added two more hallmarks and two other necessary conditions to be fulfilled [4]. 

Actually, these ten rules have become the keystone of cancer research in the last ten years and show how well organized malignant transformation and progression is. There is considerable organization behind the apparent chaos, and this organization is shared by many tumors independently of where they originate.

In the meantime, a group of researchers has been steadily investigating what we may call the eleventh hallmark of cancer: the inversion of the pH gradient—the pH paradigm.

This paradigm consists of slightly increased alkalinity of the intracellular milieu, while the extracellular substance becomes very acidic. This means that the normal pH gradient is inverted in cancer cells. Does this deserve to be considered another hallmark?

We think it does, because of its universality and the fact that proliferation, invasion, and metastatic potential are significantly reduced by reversing the inverted gradient. Furthermore, when the first step of the inversion is blocked, malignant transformation does not occur [5]. 

## 2. A Short History of the pH Paradigm

The pH paradigm in cancer is a structure that, like other concepts in science, has been built up brick-by-brick. After one hundred years of research on this subject, we now finally have a full understanding of its importance. We also know the basic mechanisms that lead to the paradigm. However, this knowledge is incomplete as yet.

Importantly, we know how to tame and reverse the paradigm. The importance of reversing the inversion of the pH gradient is looming. Animal experiments and an as yet limited number of human cases are showing the potential of this type of treatment. 

Interest in tumors’ pH stems from the pioneering works by Otto Warburg almost a hundred years ago, when he found that tumors were acidic. He was unable to distinguish between intracellular and extracellular pH at that time [6]. In 1975, Leopoldo Anghileri [7] found that inducing metabolic acidosis with ammonium chloride decreased the growth of many experimental tumors. The mechanism involved was not clarified.

In 1980, Harguindey and Gillis proposed for the first time that proton alterations in the cancer environment had a relationship with carcinogenesis and the active interference with these alterations could prevent/modify carcinogenesis [8].

The studies by Albers et al. [9] in 1981 were the first to show that there was an inversion of the pH gradient in malignant tumors. They wrote: “pH_i_ exceeded pH_e_ if either P_CO2_ and/or the concentration of lactic acid were raised above a critical level”. However, the authors expressed doubts about their own results. This publication did bring three new concepts to light: ⁂The pH gradient inversion in cancer,⁂That lactic acid was not the only cause of acidity, with CO_2_ being equally important, and⁂That pH_i_ and pH_e_ are different.

This was the starting point.

The conclusion of metabolic and physiopathologic studies on the pH gradient inversion showed that it was an essential step in cancerization and tumor progression. It should not be recognized as a hallmark if the paradigm is only a side effect of the Warburg effect [10]. It is not, because pH inversion seems to take place before the Warburg effect is functionally operative, and it is not dependent only on increased lactate production [11]. Further research is needed to determine what comes first, the Warburg effect or the pH paradigm. However, after the seminal studies by the groups headed by Pouyssegur [12,13,14,15], and Reshkin [5,16,17,18,19,20,21], evidence hints towards the second option, based on:(a)The finding of increased intracellular alkalinity at the very beginning of malignization [5].(b)The need for a high intracellular pH for the increased activity of glycolytic enzymes [22].(c)The persistence of the pH paradigm, even in conditions in which lactic acid cannot be produced.(d)The Warburg effect (aerobic glycolysis) is not indispensable for tumor survival and progression [13].(e)Hypoxia is involved in both the Warburg effect and pH inversion. Its signaling through hypoxia-inducible factors (HIFs) induces not only the overexpression of almost all the glycolytic enzymes, but the exchangers and transporters that generate the gradient inversion as well [23].(f)Intracellular pH controls growth in amniote embryos [24].

We think that both these essential hallmarks of cancer, namely the gradient inversion and aerobic glycolysis (Warburg effect), are parallel processes with many intersecting paths. HIFs are part of this network and probably, hypoxia coordinates the process as a major player. However, by definition, the Warburg effect takes place even in the presence of adequate oxygen supply, and the gradient inversion at the very beginning of cancerization occurs before any hypoxia develops. Both metabolic mechanisms form part of the adaptation of malignant cells to high proliferation, which creates harsh conditions, namely hypoxia and the increased amounts of protons. While hypoxia is handled through a form of metabolism that does not need oxygen, aerobic glycolysis, the excess of proton production is dealt with through increased proton extrusion from the cell. An excess of intracellular protons remaining inside the cell would be toxic and jeopardize the cell’s survival.

After thirty years of research on the pH paradigm, authors concur that it represents a powerful driver of cancer initiation and progression [5,25].

## 3. The pHtome and How the Paradigm Appears

pHtome is a neologism that gathers in one word all the cellular and extracellular elements that participate in creating the pH paradigm. It comprises voltage-gated channels, ion exchangers, transporters, water channels, enzymes, buffers, transcription factors, specialized areas of the cell membrane, and probably some as yet unknown proteins [26]. At least ten of all the participants in the pHtome have been identified as the paradigm’s main culprits, but this does not mean that there are no others. The pHtome participants have overlapping functions, thus inhibiting one of them has no lasting effects on the pH paradigm, since the others can easily compensate for it. Inhibition of many, on the other hand, is more difficult for the tumor to overcome. This is the theoretical framework on which the treatment scheme shown below is based.

Another important issue is that these participants coordinate among themselves to some degree. Possibly the main coordinators are the pH itself, growth factors, hypoxia, metabolic pathways, transcription factors such as hypoxia-inducible factors (HIFs) and Sp1 (specificity protein 1), and some specific genes such as Myc. We say possible coordinators, because a master coordinator or regulator has not been clearly identified beyond pH itself. The main participants of the pHtome are:(1)Sodium-hydrogen exchangers1 and 3 (NHE1 and NHE3)(2)Voltage-gated sodium channels (VGSCs)(3)Membrane carbonic anhydrases 9 and 12 (CAIX and CAXII)(4)Intracellular carbonic anhydrases(5)Sodium bicarbonate cotransporter-1 (NBC1)(6)Vacuolar ATPase proton pumps (PPs)(7)Monocarboxylate transporters 1 and 4 (MCT1 and MCT4)(8)Na^+^/K^+^/2Cl^−^ cotransporter (NKCC1)(9)Cl^−^/CO3H^−^ exchanger(10)Specificity protein 1 (Sp1) transcription factor(11)Lactate and the lactate shuttle(12)Lactate dehydrogenase(13)Growth factors(14)CO_2_(15)Aquaporins (Figure 1).

The main players in the pH paradigm: (1)Sodium-hydrogen exchangers 1 and 3 (NHE1 and NHE3)(2)Voltage-gated sodium channels (VGSCs) (Figure 1)

In the 1970s, it was discovered that sea urchin eggs underwent a significant increase of pH_i_ immediately after fertilization as a necessary condition for proliferation [27,28,29,30]. Furthermore, intracellular alkalinization can be considered a signal for proliferation [31]. This pH_i_ increase, which was also found in other species, was the result of an enhanced proton extrusion and sodium import. In 1984, Pouyssegur et al. [32] showed the critical role of NHE1 in intracellular pH regulation and the role of this regulation in cellular growth and proliferation [33]. The relationship between proliferation and migration, in cancer and NHE1, was further confirmed by other authors as well [34,35,36,37,38,39,40,41]. Furthermore, NHE1 activity also plays a role in tumor cells’ escape to the cytotoxic effects of chemotherapy [42]. Growth factors exert control on NHE1 [43]. It is probably through their signaling that the pH threshold that activates NHE1 is increased. This means that NHE1 will become active with a higher intracellular pH Figure 2 and Figure 3).

(3)Membrane carbonic anhydrases 9 and 12 (CAIX and CAXII)(4)Sodium bicarbonate cotransporter1 (NBC1) (Figure 2)

There is ample evidence showing the pro-tumoral activity of carbonic anhydrases, in particular the membrane CAs, 9 and 12 (CAIX and CAXII) [53,54,55,56,57,58,59], but also non-membrane CAIV [60], CAIII [61], and CAII.

CAs catalyze the reversible hydration of CO_2_ to form carbonic acid, which is immediately ionized to bicarbonate and a proton (CO_2_ + H_2_O → CO_3_H^+^ + H^+^) (Figure 4).

While the proton thus released contributes to extracellular acidity, the bicarbonate is reintroduced into the cell by NBC1, contributing to intracellular alkalinity.

Overexpression of membrane CAs is classically considered a sign of tumor hypoxia [62,63], however, there are tumors, such as prostate cancer, that in spite of being hypoxic, do not overexpress CAIX or CAXII [64]. 

CAs work “in tandem” with NBC1 [65] that introduces the bicarbonate produced by the CAs into the cell. The “coordinator” of this tandem work is in the hands of membrane carbonic anhydrases that regulate the amount of bicarbonate produced and at the same time, participate in the bicarbonate transport process [66,67].

Therefore, NBC1 also plays pro-tumoral roles through its association with membrane CAs or even independently [68]. Intracellular CAs also participate as facilitators of NBC1 bicarbonate transport in normal heart tissue [69] and, we suppose, in cancer as well. The exact mechanism has not been fully clarified (Figure 4). 

(5)Vacuolar ATPase proton pumps (PPs)

PPs extrude protons from the cell with energy consumption, through two mechanisms: directly or into endosomes (Figure 5). Decreasing intracellular protons increases pH_i_, while increased extrusion decreases pH_e_.

There are many publications in the medical literature showing that proton pumps play an essential role in the pH paradigm [71,72,73], and PPIs (PP inhibitors) can be a useful tool in cancer treatment [74,75,76]. There are also some controversial population studies reporting that chronic use of PPIs does not decrease cancer risk and, on the contrary, increases gastric cancer risk [77,78,79,80]. Without entering in this controversy which is based on populations that had *Helicobacter pylori* infection, and is therefore biased, it is necessary to establish that:(1)V-ATPase proton pumps play an important role in the pH paradigm.(2)They represent the main acidification machinery of endosomes whose content is released to the extracellular space or migrates from the cell as exosomes.(3)Therefore, using PP inhibitors (PPIs) as part of a pH-centered therapy responds to this logic.(4)PPIs may not prevent cancer and they may even increase the risk of gastric cancer, as the new research suggests, but the patients who receive PPIs as part of a cancer treatment already have cancer.(5)The survival time of most of these patients is usually shorter than the prolonged time PPIs administration requires for increasing the risk of a second cancer.

Therefore, our conclusion is that a possible risk increase using PPIs is inconsequential for our purposes.

(6)Monocarboxylate transporters 1 and 4 (MCT1 and MCT4)(7)Lactate (Figure 6)

For a review of both items, read the studies by Payen et al. [81] and Panisova et al. [82]. 

Monocarboxylate 4 is the main isoform for lactate transport leaving the cell. Monocarboxylate 1 introduces lactate into oxidative cancer cells in the lactate shuttle process. MCT4 and MCT1 are overexpressed in many malignancies such as non-small cell lung [83], breast [84,85], colorectal [86], gastric [87], clear cell renal carcinoma [88], and prostate [89], among many others.

The importance of MCTs is rooted in the high lactate production of almost all malignancies and thus the need to extrude it from the cell avoiding an intracellular lactic acidosis that would induce apoptosis. 

Lactate extruded from cells has been identified as a pro-tumoral factor through diverse mechanisms [90,91]. Inhibition of lactate production has shown anti-tumoral effects [92,93].

Some of lactate’s effects are related to its ability to increase extracellular acidity, however, there are pro-tumoral effects independent of the pH paradigm. 

On a theoretical basis, we have proposed increasing lactate production with metformin and at the same time, decreasing lactate extrusion by MCT4 inhibition. This situation would increase intracellular lactate with toxic effects on the malignant cell [94]. This idea has been experimentally confirmed at the cellular level [95,96,97] but has not been tested in the clinical setting. Benjamin et al. [95] used syrosingopine to achieve MCTs inhibition.

Syrosingopine is a rauwolfia derivative with antihypertensive properties [98] which has been used in clinical practice since the early 1960s [99,100,101]. It has been replaced by new antihypertensive drugs. However, syrosingopine has an interesting effect: MCT1 and MCT4 inhibition [102,103,104].

According to the authors mentioned above, syrosingopine exerts its anti-tumoral effects by depleting ATP. We think that the main action of syrosingopine is through intracellular lactic acidosis that inhibits the glycolytic flux. When used in association with metformin, this intracellular lactic acidosis is further enhanced, resulting in apoptosis.

(8)Na^+^/K^+^/2Cl^−^ cotransporter (NKCC1)(9)Cl/CO_3_H^−^ exchanger (SLC4A8) (Figure 5)

NKCC1 is a symporter that incorporates Na^+^, K^+^, and Cl^−^ to the cell. On the other hand, the chloride/bicarbonate exchanger is an antiporter that extrudes Cl^−^ and imports bicarbonate. The exchanger works in close association with CAII. Some authors consider NKCC1 expression as a potential pro-tumoral agent [105,106,107,108]. NKCC1 indirectly participates in cytoplasmic alkalinization through its association with the Cl^−^/CO_3_H^−^ exchanger [109]. Figure 7 explains the complementary mechanism of action of these two membrane proteins and cytoplasmic carbonic anhydrase II (CAII) [110,111]. The interaction between chloride/bicarbonate exchanger and CAII is so important that by inhibiting CAII with acetazolamide, the level of ion transport through the membrane decreases significantly [112].

The diuretic bumetanide inhibits NKCC1 [123,124]. However, we have not used it with patients receiving quercetin (quercetin activates NKCC1). Topiramate, which is part of the treatment scheme, seems to be able to downregulate NKCC1 [125], thus compensating quercetin’s effects on this transporter.

(10)Sp1 transcription factor

This transcription factor is involved in the regulation of many genes. Interestingly, it induces/enhances the expression of many of the participants of the pHtome, namely CAIX [126,127], NHE1 [128], NHE2 [129], NHE3 [130], NBC1 [131], and some proton pumps such as H-K-ATPase [132]. Furthermore, Sp1 transcription activity is pH-dependent [133]: low intracellular pH increases Sp1 DNA binding and its interaction with TATA binding protein. In summary, Sp1 is a housekeeper that prevents low pH_i_. Therefore, it is a player in the pH paradigm. Independently of its effects on the pH paradigm, there is abundant evidence showing the pro-tumoral activity of Sp1 [134,135,136,137].

The best known Sp1 inhibitors are tolfenamic acid [138,139] and mithramycin [140]. The pain-reliever tolfenamic acid has been safely used for the treatment of cephalea for many years, but now it is only available for veterinary use. Interestingly tolfenamic acid not only downregulates Sp1 but also has many other anti-cancer effects [141,142,143,144,145,146,147,148,149,150,151].

This short list does not mean that there are no other channels, transporters, and proteins involved, such as calcium and potassium channels, aquaporins, etc., however it shows that these above-mentioned ten are well known, have been extensively investigated, and seem to have a primary and essential role in the paradigm. Furthermore, inhibition of any of the first seven significantly modifies intra- and/or extracellular pH. However, their functions overlap and usually inhibiting just one is not sufficient.

For a long time, excessive lactate produced by increased glycolytic flux was considered as the main avenue towards extracellular acidity (recently reviewed by Parks et al. [12]). However, experimental evidence with tumors artificially engineered to become low lactate producers showed that the paradigm was still present even without this metabolite.

Our limited clinical experience has shown that targeting only one pHtome participant has little effects. We have to address at least three of them, with NHE1 being the main target.

## 4. Clinical Approach

Recognition of this eleventh hallmark of cancer raises many questions in the mind of the practicing physician:(1)Is it important?(2)Does it require treatment?(3)Can it be treated?(4)How to treat it?

(1) Is it Important?

Laboratory experiments have shown that reversing the inverted pH gradient entails great difficulties for tumor initiation [5,152], growth [153], progression [154], invasion [155], metastasis [156], and relapse [157]. The pH paradigm decreases cellular immune defenses against the tumor [158,159,160,161] and the effects of some chemotherapeutic drugs [162,163]. It also plays a role in intercellular communication through exosomes [164]. Thus, the answer is yes, it is important, because it has a fundamental role in promoting almost all the other cancer hallmarks. Furthermore, migration, invasion, and metastasis become very difficult, albeit not impossible, when the inverted gradient is returned to normality.

(2) Does it Require Treatment?

Many chemotherapeutic and targeted treatments modify the paradigm through their cytotoxic-apoptotic effects on the tumor mass. However, remaining malignant and dormant cells will reproduce the cellular and microenvironmental pH they need. It is here that targeting the paradigm may achieve better results. There is another situation in which extracellular acidity needs to be modified for a successful therapeutic result: weak basic chemotherapeutic drugs.

Interestingly, the therapeutic reversion of the inverted gradient, with no other anti-tumoral treatment, may achieve promising results, as we shall see in case Number 1.

(3) Can it be Treated?

The pH paradigm is the result of the dysregulated activity of many or all the participants of the pHtome. There are inhibitors that can partially or totally block the action of five of them. These inhibitors already exist, and many are in clinical use for other diseases. In general, they lack toxicity, or at the most, toxicity is minimal. There is no available inhibitor of NBC1 in the clinical setting. Lactate production can be decreased by inhibiting lactate dehydrogenase (LDH), however there is no effective inhibitor for bedside use. Our analysis and treatment scheme will be based exclusively on existing drugs that are suitable for treating patients.

(4) How to Treat?

The main culprit of the pH gradient inversion is NHE1. Therefore, this is the first pHtome member to be considered. However, it is necessary to inhibit many members in order to achieve anti-tumoral effects.

## 5. NHE1 and NHE3 Inhibition

Amiloride, a well-known potassium-saving diuretic that has been in clinical use for over fifty years, is an inhibitor of NHE1 and a mild NHE3 inhibitor [165,166]. Both isoforms of NHE are important for pH_i_ regulation, however there is evidence that NHE1 is the main regulator [167]. There is also evidence showing that NHE1 inhibition by amiloride has anti-tumoral effects [168,169,170,171,172], that are not limited to NHE1 inhibition, but also inhibits uPA (urokinase plasminogen activator) [173,174]. Amiloride’s anti-uPA effects seem to be of particular importance in prostate cancer where uPA’s overexpression plays an important role [175].

The usual 5 mg dose used in heart failure patients is insufficient to achieve an important inhibition of NHE1 as required in cancer treatment. A 20 mg daily dose showed minimal side effects due to hyperkalemia, which could be solved by simultaneous use of diuretics such as furosemide or hydrochlorothiazide.

Cariporide is a more potent NHE1 inhibitor compared with amiloride, although not FDA (Food and Drug Administration USA)-approved [176]. However, this drug has been tested in more than 10,000 patients in the cardiologic context (Guardian Trial) and in spite of not reaching the desired endpoints, the drug proved to be non-toxic and very well tolerated by patients [177,178,179]. In 2001, in a Doctoral Dissertation, Wong [180] was the first to call attention to cariporide as a possible drug for cancer treatment. In 2007, Sario et al. [181] found that cariporide reduced migration and proliferation of cholangiocarcinoma cells. Many reports since then have shown the anti-cancer abilities of cariporide [182,183,184,185] that confirmed these findings. In 2013, Harguindey et al. [186] suggested cariporide for cancer treatment. Subsequent publications further confirmed cariporide’s effects as an anti-cancer drug [187,188,189] and its ability to produce significant intracellular acidification in malignant cells [190]. However, cariporide is still waiting to be called into active service. Furthermore, there are no registered clinical trials for cariporide in cancer.

Other NHE1 inhibitors: There are amiloride and cariporide derivatives that are more potent than amiloride and have also shown anti-cancer properties. Not one has been approved for clinical use.

Therefore, amiloride is the only available pharmaceutical, approved by the FDA as a diuretic and available on the market.

## 6. Voltage Gated Sodium Channels (VGSCs) Inhibition

Most anti-epileptic drugs in clinical use, including phenytoin [191,192,193], carbamazepine [194,195,196,197], lamotrigine [198], topiramate [199,200,201,202], and many others, inhibit VGSCs and show different degrees of anti-tumoral effects (for a review see Koltai [203]). From a long list of VGSC inhibitors, we have chosen topiramate because it has a quadruple action against tumors:(1)It is a VGSC inhibitor [204](2)Topiramate is a potent acidifier of the intracellular milieu [205](3)It inhibits mainly CAII, and to a lesser degree CAIX and CAXII [206,207](4)It inhibits aquaporin 1 [201], aquaporin 4 [208], and aquaporin 5 [209]. Aquaporins are also participants in tumor development [210,211,212,213,214] and the pH paradigm [215,216,217]. This seems to be of particular importance in glioblastoma [218,219].

The importance of CAII inhibition by topiramate is that it has a triple effect:(a)It impedes NHE1 enhancing activation by CAII [220,221](b)It impedes intracellular recycling of bicarbonate. Both actions of topiramate reinforce its cell acidifying effects(c)It blocks other pro-tumoral effects of CAII [222,223,224,225,226,227].

The starting dose is 50 mg with weekly increments of 50 mg until achieving 200 mg daily.

NHE1 and VGSCs work together in a pro-tumoral manner, but they also have independent actions. Based on this logic, we think that NHE1 and VGSCs should be addressed simultaneously. The amiloride and topiramate association would be suitable for this purpose.

## 7. Carbonic Anhydrase (CA) Inhibition

Acetazolamide: It was the characterization of carbonic anhydrase in 1933 [228] which was followed by the discovery that sulfanilamide, the active metabolite of the sulfonamide Prontosil, inhibited CA, which led to increased natriuresis and the excretion of water [229]. Sulfanilamide gave rise to better CAs, such as acetazolamide. Diamox, the brand name of acetazolamide, was originally used for congestive heart failure and glaucoma. Nowadays, it is used for intracranial hypertension, glaucoma, altitude sickness, and Meniere’s disease. Toxicity is very low, however side effects are abundant, such as tinnitus, paresthesia, loss of appetite, vomiting, fatigue, and sleepiness. Administration of high doses of acetazolamide for long periods is not well-tolerated by patients. Therefore, Diamox should be used in the treatment of the pH paradigm only for a short time or at small doses. Some tumors, such as prostate cancer, do not show overexpression of membrane CAs, thus we have doubts about inhibiting CA in this case.

There is very active research for specific CA inhibitors to replace the unspecific acetazolamide. However, inhibiting membrane CAs is not enough to reverse the inverted pH gradient; some cytoplasmic CAs, such as CAII [222,230,231,232], CAIV [233], and other CAs [234], should be targeted too. Therefore, we believe that acetazolamide is adequate for this purpose.

## 8. V-ATPase Proton Pump Inhibition (PPI)

V-ATPase proton pump inhibition has been shown to have the ability to reduce tumor growth and proliferation [235,236]. Ferrari et al. [237] showed that PPIs increased chemosensitization in osteosarcoma patients. Wang et al. [238] reported similar findings in breast cancer patients.

There is an abundance of PPIs in clinical practice, including omeprazole, esomeprazole, pantoprazole, lansoprazole, etc., and they are all used for the treatment of gastrointestinal hyperacidity. We selected lansoprazole due to some anti-tumoral advantages such as suppression of tumor necrosis factor α (TNFα), NF-kB activity, phosphorylation of ERK [239], decreased adhesion of cancer cells to the matrix [240], synergy with metronomic chemotherapy [241], and synergy with carbonic acid inhibitors [242]. There is also evidence of pro-apoptotic activity in cancer cells [243,244], and of fatty acid synthase inhibition [245].

## 9. MCT Inhibitors

There are no FDA-approved drugs with potent MCT inhibitor abilities. However, there is a commercially available nutraceutical that can partially inhibit MCT: quercetin. Quercetin is a natural flavonoid that has shown preventive [246] and therapeutic effects in cancer:(1)Inhibiting cell cycle progression [247,248],(2)Inducing apoptosis limited to cancer cells [249],(3)Interfering with reactive oxygen species (ROS) metabolism [250],(4)Inhibiting the androgen receptor in prostate cancer [251],(5)Targeting cancer stem cells [252],(6)Downregulating mutant p53 [253] and metalloproteinases 2 and 9 [254],(7)Inhibiting p21-RAS expression [255],(8)Decreasing glycogen synthesis [256],(9)Antagonizing inflammation [257],(10)Inhibiting Hsp 27 [258],(11)Decreasing cMyc and PI3K/Akt signaling [259],(12)Decreasing lipogenesis [260],(13)Amid other anti-cancer effects [261,262].

Many of these actions are probably independent of MCT inhibition. For the purpose of the pH-centered treatment, quercetin is able to interfere with lactate export, thus decreasing intracellular pH [263,264]. This is achieved through MCT inhibition [265,266]. Decreased lactate transport diminishes fermentative glycolysis as well [267,268]. However, there is evidence showing that quercetin might increase NHE3 activation [269]. We presume that this is due to the intracellular acidification generated by quercetin and further confirms the need for simultaneous inhibition of the exchanger with amiloride when quercetin is used.

Additionally, quercetin activates the Na^+^/K^+^/2Cl^−^-cotransporter (NKCC1) [270], increasing intracellular Cl^−^. However, electroneutrality is maintained because it simultaneously imports Na^+^ and K^+^. The cytosolic acidification induced by quercetin is mainly due to MCT inhibition and lactate accumulation, and not by NKCC1 activation. There is also evidence that quercetin is able to reduce mitochondrial energy production [271]. However, this last effect requires concentrations that are difficult to achieve in the clinical setting.

One big advantage of quercetin is that neither toxicity [272,273] nor genotoxicity [274] were reported, even with very high doses [275]. Renal toxicity may ensue with doses above 1000 mg per square meter when the intravenous route is used [276]. Absorption of quercetin is not so poor as some authors have stated. The range of absorption is between 17% and 50% with an oral intake in humans. The wide range depends on the attached chemical, being the highest for quercetin glycosides and the lowest for quercetin rutinoside, while quercetin aglycone (the product found on the market) is approximately 25% [277]. The mean time to achieve maximum concentration was shorter for quercetin aglycone than the rutinoside form [278]. A dose of 330 μMoles reached a peak concentration of 5 μMoles/liter in plasma forty minutes after ingestion [279]. A single ingestion of 300 mg of quercetin achieved a plasma concentration of 9.72 μMoles [280]. A concentration of 1 μMol causes a 23% reduction of TNFα production in healthy individuals [281] and prolonged administration of quercetin significantly increased plasma concentration [282]. Therefore, we must conclude that oral administration of quercetin has an acceptable absorption and can achieve a concentration that is able to produce biologic effects.

Daily, 600 to 1000 mgs of oral quercetin is a safe dose.

Statins were proposed as MCT4 inhibitors [283,284,285]. Precisely one of the side effects of chronic statin use is its actions on muscle cells characterized by cramps due to lactic acid accumulation inside the myocytes [286].

There is no published research on the association of statins with quercetin to induce MCT inhibition. Thus, we do not know if there is synergy. Published data on this association has not investigated the issue [287]. In spite of this gap in our knowledge, we used both drugs in association in order to inhibit the MCT system.

## 10. The Role of Metformin in the pH Paradigm Treatment

The “metformin dream” began in 2005, when a large population study of diabetic patients showed that those taking metformin had a lower risk for cancer [288,289,290]. This “dream” consists of many “beliefs” about this simple and old drug, e.g., that it can:(1)Improve the results of cancer treatment,(2)Reduce the risk of cancer in non-diabetics,(3)Be included in all cancer protocols.

These beliefs have not been validated by strong clinical data. In the first place, metformin reduces the risk of cancer in diabetic patients who already have an increased risk for cancer. That result cannot be extrapolated to non-diabetic populations. Many of these supposed metformin risk reduction studies are biased and have many confounding factors [291,292] because the comparison is made with populations receiving the pro-tumoral insulin or some other anti-diabetic compound.

Adding metformin to usual cancer therapy has not shown the expected results.

However, correctly used metformin does have a place in cancer treatment, although not in risk reduction of non-diabetics. The mechanisms that have been classically postulated for metformin’s anti-tumoral action are not the most important (for the postulated classical mechanisms read Kasznicky et al. [293], Rena et al. [294], and Vancura et al. [295]). The real anti-tumoral action of metformin arises from its main mechanism of action: inhibition of Complex I of the mitochondrial electron respiratory chain [296,297,298,299,300,301,302,303]. This inhibition downregulates oxidative phosphorylation and increases glucose fermentation to lactate [304,305,306]. Lactate in the cytoplasm decreases pH_i_ [307]. Therefore, metformin is a cytoplasmic acidifier. The more glucose is metabolized, the more lactate is produced in presence of metformin. But the lactate is swiftly exported by MCT4. Metformin alone is inefficient to significantly decrease pH_i_. Metformin in high doses (3 to 4 g daily) associated with MCT inhibitors is an efficient combination to reverse the pH paradigm. This has not been tested in vivo.

## 11. The Bedside Treatment Scheme

The treatment is based on the inhibition of at least three of the elements that create the pH inversion as follows:(a)NHE1 is inhibited with amiloride.(b)V-ATPase proton pumps are treated with lansoprazole.(c)VGSCs, CA, and aquaporin 1 are targeted with topiramate.(d)Additionally, stronger inhibition of CAs can be achieved with acetazolamide. However, the tolerance for high doses of acetazolamide for long periods is very low in many patients. The main use of acetazolamide in the scheme is in the first month of treatment when topiramate has to be slowly increased. Once a full dose of topiramate is achieved, it is convenient to reduce the acetazolamide dosage. If patient tolerance is good, acetazolamide is an efficient non-specific CA inhibitor.(e)Sp1 transcription that induces HIF1 stabilization, NHE1, and CAs expressions can be inhibited with celecoxib, a COX2 inhibitor. Actually, the best drug for targeting Sp1 is tolfenamic acid (FDA-approved for human use), but it is not available in many countries.(f)MCTs are inhibited with high doses of the non-toxic nutraceutical quercetin in association with simvastatin or other statins. In a few cases, we used metformin as part of the scheme, but lately it has been eliminated due to poor patient tolerance to high doses.

The simultaneous targeting of at least three of the main players of the pHtome effectively decreases intracellular pH, thus reversing the paradigm [308].

## 12. Side Effects of the Treatment

The patients’ usual complaints are constipation, asthenia, paresthesias, dysgeusia, and orthostatic hypotension. Constipation is due to metformin and statins and responds adequately to symptomatic treatment. Paresthesias, asthenia, and dysgeusia are common effects of acetazolamide at high doses. Dose reduction decreases the symptoms but usually not completely. In some cases, acetazolamide was reduced to low doses (250 mg a day) with better tolerance. Topiramate is able to replace acetazolamide as a carbonic anhydrase inhibitor, however, acetazolamide is still needed for adequate inhibition of CAIX and CAXII. Celecoxib also has CA inhibitor abilities.

Interestingly, in patients that received the treatment for very long periods (Case Number 1), side effects subsided after a year. Orthostatic hypotension is also a common complaint. Patients should be advised to stand up slowly in order to avoid this problem. They should be instructed to have a high fluid intake to avoid kidney stone generation.

In general, the treatment was well-tolerated by all the patients. Dose adjustment was necessary in some cases.

## 13. Pet Cancer Treated with pH-Centered Schemes

Spontaneously occurring tumors in small pets—mainly cats and dogs—are closely related to human cancer and can be considered a good model of human cancer [309].

Pets, and dogs in particular, have high incidence of spontaneous cancers, and their molecular mechanisms are quite similar to those found in humans. Considering that they have a shorter life span and cancer progression time and taking into account some relative differences [310], these animal studies may be useful as a proof of concept. The group headed by Fais and Spugnini [311,312] found that the proton pump inhibitor lansoprazole was able to reverse drug resistance in 67% of dogs with spontaneous cancers, achieving complete or partial remission. Combining metronomic chemotherapy with high-dose lansoprazole, the results further improved with 75% of complete, partial, or stable disease response. Our own experience showed similar results with a full pH-centered scheme associated with metronomic chemotherapy [313].

## 14. Results of pH-Centered Treatments in Humans

Eleven patients with different, very advanced tumors, were treated on a compassionate basis with part or the total scheme. Ten patients were under palliative care and were not receiving chemo- or radiotherapy at the time they started the treatment. They were all heavily treated previously, and the disease was in progression with multidrug resistance.

Six patients did not show any improvement with the scheme. In this group of non-responders, there was one with relapsing glioblastoma, two patients with pancreatic cancer, one case of ovarian metastatic cancer, one patient with colon cancer, and one with gastric adenocarcinoma.

Five patients showed stable disease for prolonged periods:

One patient with metastatic renal cell carcinoma (Case Number 1) achieved stable disease for the last 68 months and continues without signs of progression up to now.

One patient with relapsing gastric lymphoma has not shown signs of progression after 48 months (Case Number 2).

Of the two patients with prostate carcinoma, one showed stable disease for 19 months before progression reappeared (Case Number 4). The other patient has been stable for 48 months (Case Number 3). In Case Number 4, the patient was simultaneously treated with a metronomic anti-angiogenic scheme based on daily oral cyclophosphamide and fenofibrate. Case Number 5, a patient with wide-spread metastatic melanoma, achieved stable disease for 11 weeks.

All the patients signed an informed consent after a detailed explanation of the treatment. This consent was co-signed by two witnesses, being at least one of them a family member or close friend of the patient. 

No authority permissions were required for the treatment because all the drugs and nutraceuticals used, were approved by the National Drug and Medical Technology Administration of Argentina (ANMAT) and no experimental or non-approved drugs were administered to the patients.

## 15. Human Case Presentation

Case 1: In 2002, an asymptomatic 43-year-old male was diagnosed with a right renal cell carcinoma by chance during an abdominal X-ray checkup. A radical right nephrectomy was performed, and a 3.5 cm diameter clear cell renal cell carcinoma was found and removed. There were no regional or distant metastases at the time.

Three years later, three nodular metastases of less than one cm appeared in the right lung and a mediastinal metastasis was detected shortly thereafter. All the metastases were surgically removed. The patient remained asymptomatic for two years, when two spine and one iliac bone metastases appeared. He received radiotherapy and seven months of treatment with daily 25 mg of sunitinib (2013). He achieved stable disease but developed severe heart failure and hypertension. In 2014, he showed a rib metastasis and two small left lung metastases (less than 1 cm each). In 2014, he started with the pH gradient inversion treatment and cardiologic drugs. The treatment consisted of lansoprazole 180 mg, amiloride-base diuretics (5 mg amiloride plus 25 mg hydrochlorotiazide three times a day), acetazolamide 250 mg, celecoxib 400 mg, quercetin 600 mg, and silymarin compound. The acetazolamide was poorly tolerated, and it was replaced with topiramate 200 mg (started with 50 mg and increased 50 mg every ten days). The patient achieved stable disease. The metastases did not disappear, but they did not grow any further, as his last control in March 2020 showed. Tolerance for the treatment was and is excellent, with the main adverse effects being constipation, slight hypotension, and minor asthenia. Constipation was treated symptomatically. During the six years he received the treatment, he was able to return to a normal lifestyle, including a divorce, a remarriage, and having a baby.

The scheme used in Case 1 is shown in Table 1.

Figure 8 and Figure 9 show bone surveys of the patient.

The patient is now 61 years old and has stable disease which has not evolved over the past 6 years.

Case 2: An 84-year-old woman presented a gastric lymphoma in 2014. She was treated only with omeprazole and rituximab. No other drugs were used due to the patient’s poor conditions. Complete response was achieved for four years. In 2018, an important gastric bleeding led to a gastroscopy that showed tumor relapse. The patient and the family rejected any aggressive treatment or chemotherapy, thus, a pH-centered therapy (lansoprazole, quercetin, metformin, amiloride, and silibinin in similar doses as in Case Number 1) was started. The patient’s physical conditions improved, and a control gastroscopy performed in 2020 showed the presence of lymphoma cells in the biopsy without evidence of tumor growth. Computed axial tomography showed no other organ involvement. Follow-up of 26 months.

Case 3: This 66-year-old patient is the exception in the sense that he was not a terminal case. The patient had a localized prostate cancer with no detectable metastasis. He was treated with radiotherapy and refused hormonal or chemotherapy treatments. Two years after the radiotherapy, persistent and progressive elevation of PSA ensued. No metastases were detected. The patient was started with the pH-centered treatment and PSA progressively diminished, achieving normal levels after one year. Six years after the initial diagnosis, the patient shows no metastasis and PSA is within normal limits.

Case 4: A 59-year-old patient was started on the pH-centered treatment in association with metronomic chemotherapy in 2010 for a metastatic castration-resistant prostate cancer after progression with radical prostatectomy, hormonotherapy, and chemotherapy. Bone metastases, a total of 12 (ribs, spine, pelvis, and femur), were treated with radiotherapy. The pH-centered treatment (lansoprazole, acetazolamide, silymarin, and riluzole) with metronomic chemotherapy (cyclophosphamide, celecoxib, and bazedoxifene) was initiated when the patient was on palliative care. There were neither new metastases nor growth of the existing ones for 19 months. After this period, two new bone metastases appeared and there was no further response to treatment.

Case 5: A 29-year-old male was started on a pH-centered metronomic chemotherapy scheme for the treatment of metastatic melanoma, with pulmonary, liver, spleen, adrenal, lymph nodes, and brain metastases, unresponsive to immune checkpoint and BRAF inhibitors. There were more than 20 metastases in the liver and lungs. There were two small brain metastases without cranial hypertension syndrome. The patient showed stable disease for eleven weeks, after which new metastases appeared.

## 16. Discussion

To the best of our knowledge, this is the first publication of the results obtained with a “cocktail” of proton extruder inhibitors in patients with very advanced stages of cancer.

There are two shared issues in the four cases presented here (Case Number 3 is excluded):(1)Above average survival.(2)Stable disease achieved with pH gradient inversion treatment.

Of course, there have been blatant failures too: the other six cases. However, all six failures occurred with patients almost in extremis (all of them passed away within three months). Therefore, an important question remains unanswered: would the results be improved with an earlier pH-centered treatment? Only further research can answer that.

pH-centered treatments also achieved stable disease in spontaneous dog and cat cancers.

This limited clinical experience shows that in some cases, pH treatments can achieve stable disease and decrease metastasis, although no complete or partial remission is within its abilities. Case Number 1 is a good example that with stable disease, it is possible to live a normal and good-quality life. Case Number 5 only shows a short period of stable disease, but this was brought about by a pH-centered treatment after all known therapeutic measures failed and in a very advanced stage of the disease, where remissions or stabilization are unusual.

The small number of cases, the diversity of the tumors, and the presence of confounding factors, does not permit the establishment of a proof of concept. However, a scheme of treatment was proposed which is rationally based on translational and pharmacological fundamentals. The concept of a proton extruder inhibitor cocktail as proposed by many authors has been materialized here with repurposed drugs and nutraceuticals. This is only a first step. Improving the cocktail and performing clinical trials should follow.

Quiescence of cancer cells is a major problem in tumor targeting because these cells are not responsive to classical therapies [334]. This situation is frequently a consequence of chemo- or radiotherapy [335]. Interestingly, quiescent cells have a low intracellular pH and low expression of NHE1 [336]. Our interpretation of the clinical results of the pH-centered treatments is that it is able to induce a state of quiescence in cancer cells (stable disease) rather than apoptosis (remission). The unknown issue is how to prolong this quiescence.

## 17. Future Perspectives

For bioethical reasons, we were constrained to use only FDA-approved drugs or nutraceuticals, even though they were approved for other uses than cancer. These important limitations meant that we had to leave aside some better drugs. For example, amiloride is a weak NHE1 inhibitor in the clinically achievable concentrations. Cariporide is significantly more potent, and in spite of its low toxicity, it is not an approved drug. Another example is that of MCTs inhibition with quercetin. It is a nutraceutical, and it is available over the counter on the market. On the other hand, the old anti-hypertensive drug syrosingopine is no longer available for clinical use. Something similar occurs with tolfenamic acid, probably the best non-toxic, and well-known inhibitor of Sp1 and Sp3, which is only available for veterinarian use.

Improving the pHtome inhibition with new and old drugs may perhaps achieve better results with the pH-centered treatment. Hopefully, some old drugs may make a comeback, namely tolfenamic acid and syrosingopine. Hopefully, also, maybe in the future, quercetin will receive the attention it deserves, not as a nutraceutical but as a drug with clear and confirmed anti-cancer properties.

The second issue is that no clinical trials are being conducted on a pH-centered scheme. If the future brings a well-planned clinical trial and awakens awareness about this approach, the gap between the experimental laboratory and bedside therapy may narrow.

## 18. Conclusions

The last 30 years have witnessed growing interest and research on the tumor microenvironment. This research has shown that tumor acidity, as originally conceived by Warburg and his contemporaries as a by-product of excessive glycolysis and lactate production, was the tree hiding in the forest. The investigation of the forest, rather than the tree, showed a complex world in which intracellular alkalinization, pH gradient inversion, the mechanisms involved, and the consequences thus produced became a whole new chapter in cancer biology. However, this knowledge has not reached bedside medicine. After briefly reviewing the pathogenesis of the pH paradigm, this contribution shows some practical ways of addressing the issue outside the laboratory. What we have learned from basic research and from a very limited clinical experience is that:(1)pH gradient inversion, known as the pH paradigm, is a hallmark of cancer.(2)This hallmark is not the consequence of metabolic changes, but a parallel process that may even precede the metabolic switch.(3)However, it is not independent from the metabolic switch. The latter has the ability to further increase the pH gradient inversion and at the same time, requires increased intracellular pH to enhance the activity of some glycolytic enzymes.(4)Metabolic switch and pH gradient inversion let the cell acquire proliferative and growth advantages.(5)Reversing the pH paradigm decreases growth, proliferation, and tumor progression in the experimental laboratory level and also at the bedside.(6)Reversal of the pH paradigm can be achieved with existing drugs by repurposing them.(7)The aim of the treatment is based on the simultaneous inhibition of many (at least three) of the components of the pHtome.(8)It is insufficient to block one or two proton extruders. As many as possible proton extrusion mechanisms need to be inhibited in order to get significant results.(9)NHE1 should always be one of the targets in any pH-centered scheme.(10)The treatment scheme hereby shown is only an example that can be improved with better inhibitors of the pHtome.(11)The cases reported in this paper do not represent a proof of concept. Many confounding factors could not be ruled out and the population was too small to achieve statistical significance.(12)There were no complete or partial remissions. However, stable disease for an extended period is an achievable result.(13)Resistance to pH-centered treatments may be present from the beginning (failed cases) or may develop at variable times, or almost never (Cases 1 and 2).(14)How and why resistance to pH-centered treatments develops has not been investigated as yet. Tumor heterogeneity and flexibility to changes is probably at the roots of resistance.(15) pH-centeredcancer treatment deserves well-planned large-population studies, with reliable statistics and long follow-up.

We hope this article will encourage fellow colleagues to follow the research and mainly to consider pH-centered treatments worthy of serious clinical trials. Until the results of these trials are available, “the defense rests”.

## Figures and Tables

**Figure 1 ijms-21-09221-f001:**
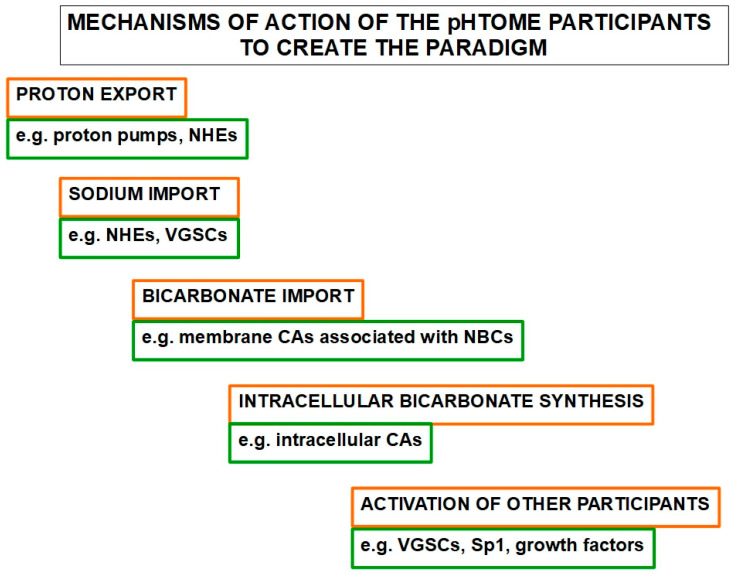
Mechanisms of action of the pHtome participants.

**Figure 2 ijms-21-09221-f002:**
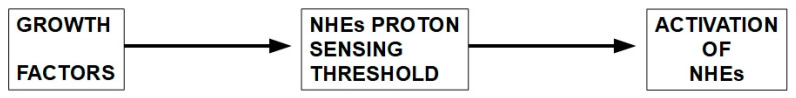
Mechanism of pH threshold modification of NHE1.

**Figure 3 ijms-21-09221-f003:**
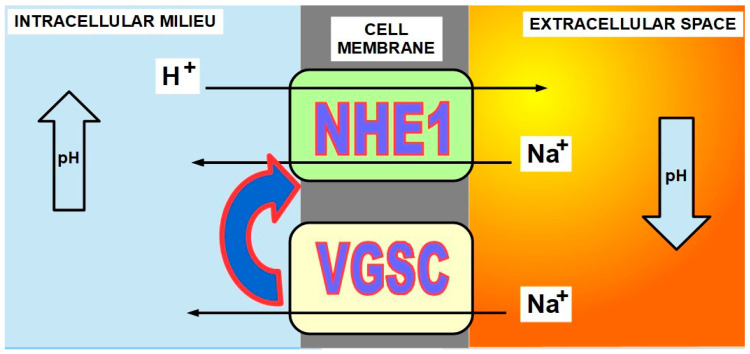
Mechanism of action of NHE1 and VGSCs. NHE1 is an exchanger that imports sodium ions while exporting protons (hydrogen ions). This proton removal from the cell increases intracellular pH while it increases in the extracellular space. VGSCs import sodium ions but do not export protons. Among their other effects, the importance of VGSCs lies in the fact that they activate NHE1 [44]. Activating NHE1 means that it “starts working” at a higher intracellular pH. Under normal conditions, NHE1 has a pH_i_ threshold and becomes active when pH_i_ goes below it. VGSCs increase the threshold, leading to NHE activation even under higher intracellular pH. There is a well-established relationship among growth factors, NHE1, intracellular increase of pH, and proliferation [35,45,46,47,48,49]. NHEs mediate the signaling of growth factors and proton sensing [50,51,52] (Box 2).

**Figure 4 ijms-21-09221-f004:**
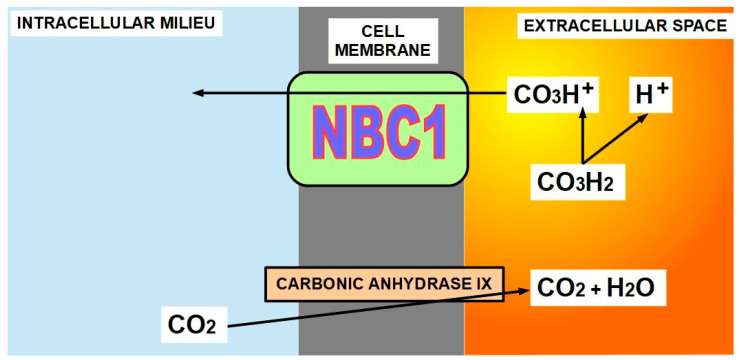
Membrane carbonic anhydrase IX generates carbonic acid through hydration of CO_2_. Carbon dioxide diffuses from inside the cell and is a product of cellular metabolism. Carbonic acid is immediately ionized to bicarbonate and a proton. While the proton remains in the extracellular space contributing to its acidity, the bicarbonate ion is imported into the cell by NBC1 contributing to increasing pH_i_. CAIX (carbonic anhydrase 9) and CAXII (carbonic anhydrase 12) have been found to be potent drivers of cancer growth by alkalinizing the intracellular milieu [70].

**Figure 5 ijms-21-09221-f005:**
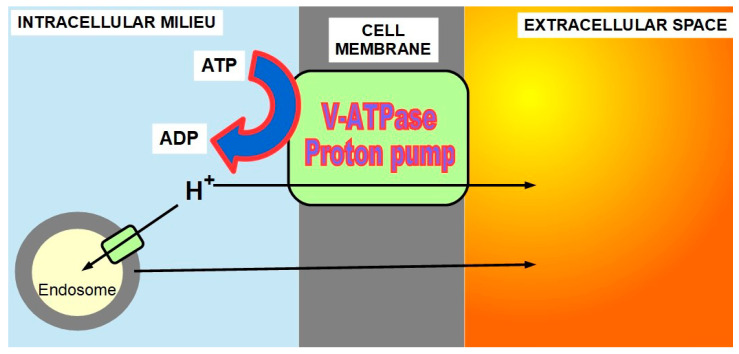
V-ATPase proton pump mechanism of action.

**Figure 6 ijms-21-09221-f006:**
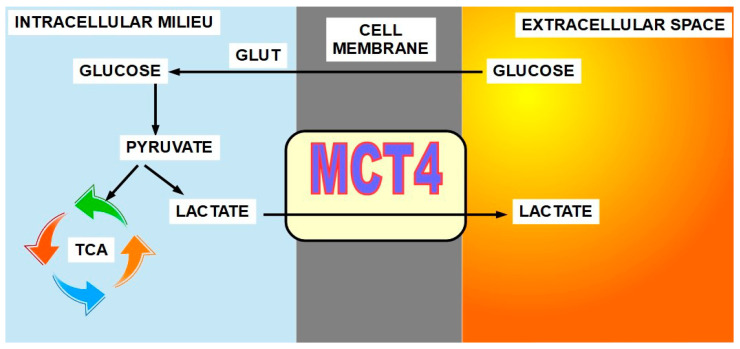
Lactate extruder function of MCT4. Lactate originates from the enzymatic glycolysis of glucose introduced from the extracellular space with the mediation of glucose transporters (GLUTs). MCT4 is the main lactate exporter, while MCT1 imports lactate into the oxidative cells participating in the lactate shuttle.

**Figure 7 ijms-21-09221-f007:**
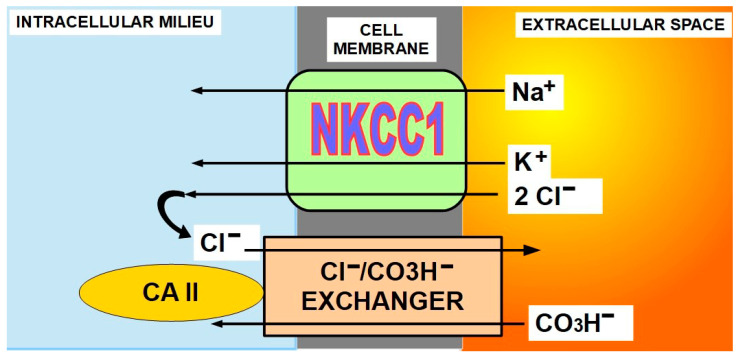
The tandem work of NKCC1 and the Cl/CO3H exchanger alkalinize the intracellular milieu. Loop diuretics in general have the ability to inhibit NKCC1. The energy required for NKCC1 is provided by the electrochemical gradient of sodium [113,114,115,116,117]. The pH_i_ stabilizing properties of the chloride/bicarbonate exchanger have been clearly demonstrated in osteoclasts [118] and kidneys. In cancer, it works in close association with cytoplasmic carbonic anhydrase II [119], with a final result of proton extrusion [120]. While NKCC1 increases intracellular Cl^−^, this molecule is re-exported by the exchanger, allowing the import of bicarbonate. Bicarbonate transport in cancer is a pivotal issue in the pH paradigm [121]. Disulfiram, a drug used to treat alcoholism, has been shown to decrease the activity of the chloride/bicarbonate exchanger [122].

**Figure 8 ijms-21-09221-f008:**
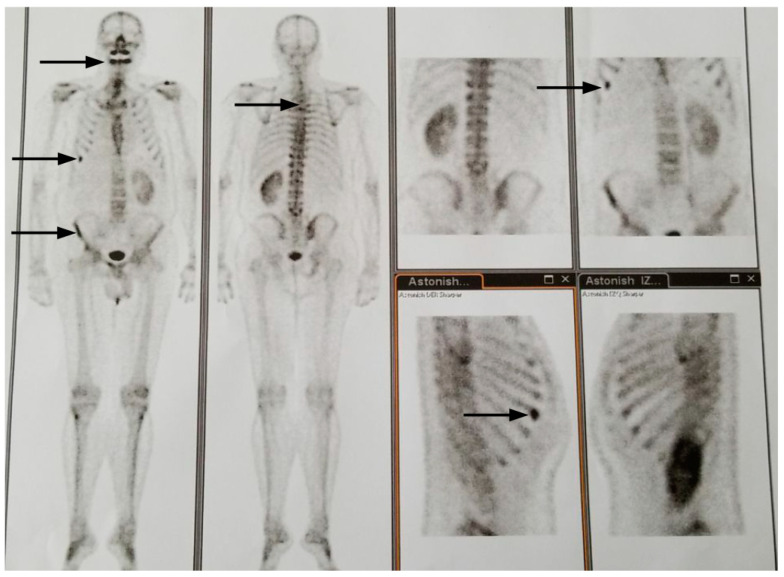
Year 2017: Front view: rib metastasis, iliac bone metastasis, increased radionuclide caption at maxillary alveolar level. Dorsal view: 4–5 dorsal vertebra metastasis. These images were similar to those of 2014. Absence of right kidney due to radical nephrectomy in 2003. Left panel anterior view. Second panel posterior view. Four right panels: upper left: posterior view, upper right: anterior view, lower left: right ribs, lower right: left ribs.

**Figure 9 ijms-21-09221-f009:**
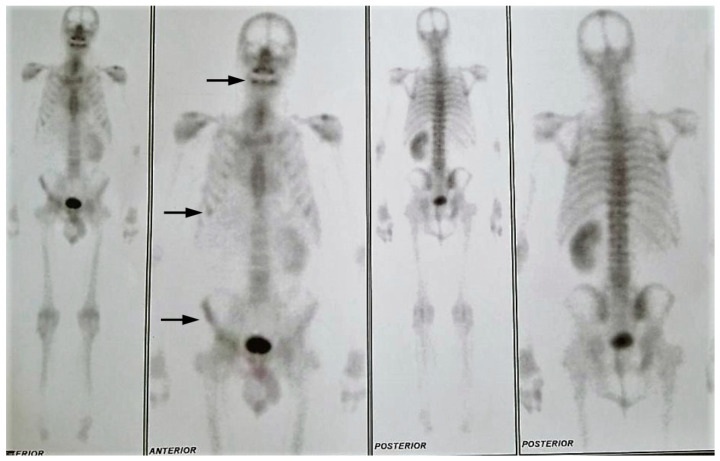
Year 2018: Front view: rib metastasis, iliac bone metastasis, increased radionuclide caption at maxillary alveolar level. Stable disease.

**Table 1 ijms-21-09221-t001:** Drug scheme used in case 1.

DRUG	DOSE	NOTE
Amiloride + Furosemide	5 mg + 40 mg three times a day.	This product is available on the market. The combination of Amiloride + Dihydrochlorotiazide can also be used. Potassium level control is required.
Lansoprazole	60 mg three times a day.	Other omeprazole derivatives can be used, namely esomeprazole, pantoprazole, etc. Our preference has been lansoprazole for the reasons explained above.
Topiramate	200 mg daily.	Starting dose 50 mg with weekly increase of 50 mg until reaching 200 mg
Atorvastatin	15 mg twice a day.	
Quercetin	100 mg three times a day.	Over the counter product. Higher doses can be used without evidence of toxicity.
Metformin	2 g a day	Tolerance was low and it was finally reduced to 500 mg a day.
Acetazolamide	250 mg twice a day.	It was discontinued due to poor tolerance.
Silymarin * (milk thistle) and silibinin	Normalized extract 200 mg twice a day	Silymarin extracts are MCT inhibitors [314] and have shown anti-tumor activity in renal cell carcinoma [315,316,317,318,319,320,321,322,323,324,325,326,327,328] and in other tumors as well [329,330,331,332].
Celecoxib + aspirin **	200 mg a day + 300 mg	Aspirin was added to decrease cardiovascular risks of celecoxib.

* Most of the anti-tumoral actions attributed to silymarin ingredients, like silibinin, are probably cytoplasm-acidification-independent. For a review on silibinin anti-tumoral effects, read Boojar et al. [333]. ** Aspirin is used as a preventive for celecoxib adverse events.

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
