# Peer review of "Targeting the pH Paradigm at the Bedside: A Practical Approach"

_ijms, 2020, doi:10.3390/ijms21239221_

Round 1

Reviewer 1 Report

I found this review/case report to be very well present and written. The extensive Introduction/review prepared the reader to understand the case that was reported. The figures and explaination of the case report were clear and relevant. Lasly, this topic is extremly important.

The only problem that needs to be corrected by the journal is that the first set of Boxes and Figures were not completely observable as they were probably too big. However I was able to understand them and see that were complete and added to the text.

I recommend immediate acceptance.

Author Response

Dear Reviewer 1

The spell check and eventually the syntax will be further revised/corrected by a native English speaker. The figures will be improved. Probably, you could not adequately visualize figures and boxes for reasons related to what the Publisher sent you. I cannot make improvements on that issue. At least this is what seems to have also happened with Reviewer 2.

Kind regards

Tomas Koltai

Reviewer 2 Report

Remarks to the Author:

Congratulations for this nice manuscript, this is an impressive and comprehensive work which contributes to synthetize the variety of mechanisms involved of the establishment of the pH paradigm and highlight the therapeutic potential of these different targets to improve cancer therapies.

The submitted manuscript is a rationale update of previous works from this author who have already published a significant number of reviews on this pH paradigm theme.

In the present case report, Dr. Tomas Koltaï thus use his experience as a clinician to go beyond the theory and therefore propose a bedside treatment scheme using a combination of repurposed drugs. Dr. Koltaï then presents a few human cases treated on a compassionate basis with just a part or the complete therapeutic scheme to illustrate the potential benefits that could be expected from the proposed strategy.

The obtained data and interpretation in the context of cancer treatment are quite original and valuable, even if these many considerations have already been discussed in the last few years, including a number of reviews in International Journal of Molecular Sciences, notably by Salvador Harguindey and collaborators.

Drafting quality and structure of the manuscript are correct, even if they require a few improvements to come up to the required standards, especially for the BOX contents and the figures. English require a few editing to improve  the clarity and the readability of the manuscript.

Introduction is rather very well set up to raise the issue and presents the many potential targets to counteract pH paradigm in the context of cancer treatment.

Although empirical, the overall therapy design seems to be adequate to the aims of the study and complies with generally accepted deregulations regarding current pH paradigm understandings. However, the presented case reports are quite poor. As specify by the author himself, the few patients included, the diversity of tumors and the presence of confounding factors make it difficult to conclude about these clinical outcomes, and ultimately about the potential benefices of the proposed treatments. Moreover, biological information’s on patients, also the chronology in patient care, are lacking to really conclude on the effectiveness of the treatments involved. As things stand at present, we cannot even precisely identify the specific treatments received by each patient. I would recommend to greatly improve the presentation of these different case reports to ensure compliance with international standards.

Regarding these last comments, perhaps the content could be published as a review rather than a case report?

I can therefore suggest to accept this manuscript entitled “Targeting the pH Paradigm at the Bedside: A Practical Approach” for publication in International Journal of molecular Sciences after a few additions and modifications.

Specific comments:

Abstract, pp.1, line 26 and 31: I would replace “will” by “would be able to” to make it understand that the proof of concept has not been validated yet.

Introduction, pp.2, line 70-72: There is a repetition dealing with what Otto Warburg and pHe/pHi.

Suggestions: I would also discuss about data on environmental carcinogenesis in this part. It is a supplementary proof of the importance of these many and well-discussed dysregulations in the ethiopathogenesis of cancers.

Introduction, pp.3, line 111-114: this part is very confusing. I think it could be easily improved.

The pHtome […], pp.3, line 132: Please add  ”:” after “the pHtome are” and add a space between this sentence and the following list. The 1) starting line 130 is thus confusing with the following numbering.

BOX1, pp.4: Presentation of BOX1 is a bit rough and does not add much to the manuscript.

Idem for BOX 2 pp.4-5 : scheme is absolutely not readable, zoomed and split between page 4 and 5. I was simply not able to rate this content !

Some modifications have to be made to increase the overall quality of the figures:

Figure 1, p.5: same kind of problems than for BOX 1 and 2. Legend of figure 1 must be improved. Presentation of the crosstalk between NHE transporter and VGSC could easily be improved.

Figure 2, 3 and 4: I think the problems come from the editor, but these figures are not readable on the version of the manuscript I received. Not easy to review in this context but these simple diagrams will undoubtedly allow non-specialist clinicians to easily follow the concepts presented by the author.

Author Response

However, the presented case reports are quite poor. As specify by the author himself, the few patients included, the diversity of tumors and the presence of confounding factors make it difficult to conclude about these clinical outcomes, and ultimately about the potential benefices of the proposed treatments. Moreover, biological information’s on patients, also the chronology in patient care, are lacking to really conclude on the effectiveness of the treatments involved. As things stand at present, we cannot even precisely identify the specific treatments received by each patient. I would recommend to greatly improve the presentation of these different case reports to ensure compliance with international standards.

Regarding these last comments, perhaps the content could be published as a review rather than a case report?

The Reviewer is right. The manuscript does not fulfill the usual standards of a Case Report. Therefore, it can be considered a review with clinical examples, rather than a Case Report. However, the clinical data in the manuscript and the treatment in each case will be further improved. A full history on each case would make the manuscript excessively lengthy and we tried to avoid that.   

I can therefore suggest to accept this manuscript entitled “Targeting the pH Paradigm at the Bedside: A Practical Approach” for publication in International Journal of molecular Sciences after a few additions and modifications.

Specific comments:

Abstract, pp.1, line 26 and 31: I would replace “will” by “would be able to” to make it understand that the proof of concept has not been validated yet.

I agree on the corrections. It will be done.

Introduction, pp.2, line 70-72: There is a repetition dealing with what Otto Warburg and pHe/pHi.

It will be corrected in order to avoid redundancy. Line 72 was deleted.

Suggestions: I would also discuss about data on environmental carcinogenesis in this part. It is a supplementary proof of the importance of these many and well-discussed dysregulations in the ethiopathogenesis of cancers.

On this point, I think that environmental carcinogenesis and pH is a completely separate issue which does not modify the core of the manuscript and it is not the main purpose of the presentation. This paper is about a practical method to treat the pH paradigm and cancer and not about the ethiology and pathogenesis of tumors.

Introduction, pp.3, line 111-114: this part is very confusing. I think it could be easily improved.

I think the sentence is very clear and it does not need any improvement.

The pHtome […], pp.3, line 132: Please add  ”:” after “the pHtome are” and add a space between this sentence and the following list. The 1) starting line 130 is thus confusing with the following numbering.

Done

BOX1, pp.4: Presentation of BOX1 is a bit rough and does not add much to the manuscript.

BOX 1 is addressed to readers who are not very familiar with the pH paradigm. It is true that it is simple, in any case, but not rough. I strongly believe it should remain in the paper.

Idem for BOX 2 pp.4-5 : scheme is absolutely not readable, zoomed and split between page 4 and 5. I was simply not able to rate this content !

This is probably a problem of the copy sent to the Reviewer. I cannot do anything about that.

Some modifications have to be made to increase the overall quality of the figures:

Figure 1, p.5: same kind of problems than for BOX 1 and 2. Legend of figure 1 must be improved. Presentation of the crosstalk between NHE transporter and VGSC could easily be improved.

BOX 2 and Figure 1 are a unity in the sense that BOX 2 helps understanding figure 1. Crosstalk between NHE1 and VGSCs is mentioned and referenced in the legend. Going further on the issue would mean a long explanation that goes beyond the scope of the manuscript. The purpose of the paper is to introduce the reader on the fundamentals of the treatment. Entering into more details at the molecular level would only confuse the reader about the essentials.

Figure 2, 3 and 4: I think the problems come from the editor, but these figures are not readable on the version of the manuscript I received. Not easy to review in this context but these simple diagrams will undoubtedly allow non-specialist clinicians to easily follow the concepts presented by the author.

This is probably a problem of the copy sent to the Reviewer.